# Iron(III)-Complexes with *N*-Phenylpyrazole-Based Ligands

**Tanja Hirschhausen** [1,†], **Lorena Fritsch** [1,†], **Franziska Lux** [1], **Jakob Steube** [1], **Roland Schoch** [1], **Adam Neuba** [1], **Hans Egold** [1] and **Matthias Bauer** [1,2,*]

1 Institute of Inorganic Chemistry, Paderborn University, 33098 Paderborn, Germany;
tanja.hirschhausen@upb.de (T.H.); lorena.fritsch@upb.de (L.F.); lux.franziska@web.de (F.L.);
jakob.steube@upb.de (J.S.); roland.schoch@upb.de (R.S.); adam.neuba@upb.de (A.N.);
hans.egold@upb.de (H.E.)
2 Center for Sustainable Systems Design, Paderborn University, 33100 Paderborn, Germany
* Correspondence: matthias.bauer@upb.de
† These authors contributed equally to this work.

**Abstract:** The use of iron as a replacement for noble metals in photochemical and photophysical applications is challenging due to the typically fast deactivation of short-lived catalytically active states. Recent success of a cyclometalated iron(III) complex utilizing a bis-tridentate ligand motif inspired the use of phenyl-1*H*-pyrazole as a bidentate ligand. Five complexes using the tris(1-phenylpyrazolato-*N*,C$^2$)iron(III) complex scaffold are presented. In addition to the parent complex, four derivatives with functionalization in the meta-position of the phenyl ring are thoroughly investigated by single crystal diffractometry, UV-Vis-spectroscopy, and cyclic voltammetry. Advanced X-ray spectroscopy in the form of X-ray absorption and emission spectroscopy allows unique insights into the electronic structure as well as DFT calculations. The ligand design leads to overlapping MLCT and LMCT absorption bands, and emissive behavior is suppressed by low-lying MC states.

**Keywords:** photosensitizer; iron(III) complex; cyclometalation; phenyl-1*H*-pyrazol

## 1. Introduction

Noble metal complexes based on ruthenium(II) [1,2], osmium(II) [3], and iridium(III) [4,5] show a long history of photophysical and photochemical applications due to their stability and activity. However, the scarcity and high costs of noble metals prevent a decentral application in water splitting or photocatalysis [6]. It may therefore be appropriate to shift the focus towards more abundant, less expensive, and, at best, more environmentally friendly alternatives. Iron is the dream candidate to fulfill these requirements. However, its use in photoactive complexes requires the development of new ligand designs for catalytically active states, since deactivation by the rapid population of inactive states usually occurs [7,8]. If this major problem can be solved, iron complexes could enable photocatalytic reactions through high-energy states with sufficiently long lifetimes. In d$^6$ systems, this is typically a metal-to-ligand charge transfer (MLCT) state [9]. These are the active states in octahedral noble metal complexes with polypyridyl-based ligands, such as [Ru(tpy)$_2$]$^{2+}$ (tpy = 2,6-Bis(2-pyridyl)pyridine) and [Ru(bpy)$_3$]$^{2+}$ (bpy = 2,2′-bipyridinyl) [10–12]. Analogous nitrogen-coordinated iron(II)-d$^6$ complexes suffer from short MLCT lifetimes in the 100 fs range due to fast relaxation into low-lying metal-centered (MC) states induced by the inherent small ligand field splitting of 3d-metals [13,14].

Consequently, strategies in ligand design target the stabilization of photoactive long-lived charge transfer states and the destabilization of MC states. Various attempts to achieve this goal employ a mix of strong σ-donor and π-acceptor ligands [14]. Strong σ-donating groups promote higher ligand field splitting by destabilization of the antibonding e$_g$*. In this context, *N*-heterocyclic carbenes (NHCs) were extensively applied due to their strong σ-donating character [15]. Starting from the bis-tridentate prototype NHC iron(II) complex

[Fe(pbmi)$_2$]$^{2+}$ (pbmi = 2,6-bis(imidazol-2-ylidene)pyridine), donor/acceptor properties could be modified by the introduction of different functional groups [14,16,17]. Nevertheless, the MLCT lifetimes of these complexes remain below 50 ps; therefore, a breakthrough using CˆNˆC ligands could barely be achieved [8].

In contrast, the substitution of the central pyridine in the CˆNˆC ligand scaffold by a phenylene to obtain a CˆCˆC ligand significantly alters the properties of the resulting complex. The cyclometalated bis-tridentate iron(III) complex [Fe(ImP)$_2$][PF$_6$] (HImP = 1,1′-(1,3-phenylene)bis(3-methyl-1-imidazol-2-ylidene)) exhibits dual emission from ligand-to-metal charge transfer ($^2$LMCT) with a lifetime of 240 ps and a $^2$MLCT state lifetime of over 4 ns [18]. In general, iron(III) complexes in a low-spin $^2$T$_2$-ground state and a strong donor environment lead to the population of $^2$LMCT states, from which spin-allowed luminescence occurs [19,20]. $^{4/6}$MC states could interfere in this deactivation pathway if they are energetically favored [21,22].

Despite recent progress using bis-tridentate coordination environments, bidentate ligands offer complexes with higher symmetry, resulting in a stronger ligand field [8,19,23]. Accordingly, in Ru(II) complexes for example, a comparison of [Ru(tpy)$_2$]$^{2+}$ and [Ru(bpy)$_3$]$^{2+}$ reveals a prolonged and much more intense luminescence at room temperature for the bipyridine complex [24]. This is reflected in [Fe(btz)$_3$]$^{2+}$ (btz = 3,3′-dimethyl-1,1′-bis(p-tolyl)-4,4′-bis(1,2,3-triazol-5-ylidene)), a hexa-carbene iron(II) complex with a $^3$MLCT lifetime of 528 ps, which is an order of magnitude longer than the lifetime of previously discussed bis-tridentate CˆNˆC complexes [25]. Its iron(III) congener [Fe(btz)$_3$]$^{3+}$ was the first iron complex to show the aforementioned $^2$LMCT emission, with a lifetime of 100 ps [19].

Consequently, the transfer of the cyclometalation approach to bidentate ligands suggests phenylpyrazole (ppz) as one possible ligand. It was already successfully employed in iridium complexes, resulting in Ir(ppz)$_3$ [26]. Although the base–metal complexes Co(ppz)$_3$ and Fe(ppz)$_3$ have also been reported, no spectroscopic analyses were conducted for the latter [27,28]. Bridging this gap by revisiting this pristine complex and further investigating its functionalization in the meta-position of the phenyl ring relative to the iron center with electron-withdrawing trifluoromethyl groups and electron-donating methoxy groups, as well as the extension of the aromatic system with phenyl and naphthalene, is the key element of this study.

## 2. Results

### 2.1. Synthesis and Characterization

The ligands were synthesized as reported in the literature by a reaction between the functionalized phenyl-bromide and pyrazole in a copper-catalyzed *N*-arylation under mild conditions, resulting in high-yield product formation (>90%) [29,30]. The synthesis of the complexes is based on the literature and involves the two steps of orthometalation and transmetalation [27,28,31]. The orthometalated ligands were obtained by refluxing the proligand with an ethylmagnesium bromide solution (EtMgBr in THF) for 24 h in THF. FeBr$_2$(THF)$_{1.5}$ was used as the iron precursor, which was prepared in situ as described in the literature and added to the ligand solution at −80 °C, thereby initiating the transmetalation [32]. The synthesis is displayed in Figure 1 (top), exemplarily for tris(2-phenylpyrazolato-*N*,*C*$^2$)iron(III) (**Fe(ppz)$_3$**). Functionalized derivatives are also shown in Figure 1 (bottom).

Workup under atmospheric conditions with an aqueous ammonium chloride solution removed bromide-containing by-products and was followed by column chromatography. After isolation of the yellow (**Fe(ppz)$_3$, Fe(CF$_3$ppz)$_3$**) or red (**Fe(bppz)$_3$, Fe(naphpz)$_3$, Fe(MeOppz)$_3$**) solids, the air- and moisture-stable products were dried under vacuum and obtained in elemental-analysis purity, albeit in low yields (<15%). These can be explained firstly by the disproportion mechanism leading to the product formation and secondly by various, unstable coordination products apart from the desired complexes [26–28].

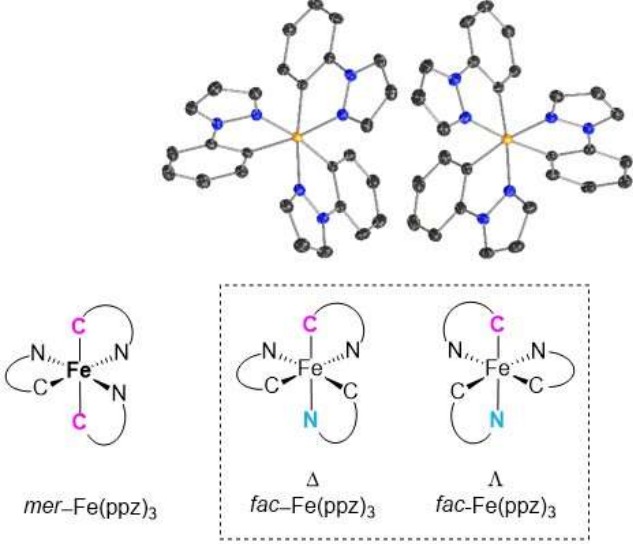

**Figure 1.** (**Top**) Reaction pathway for the synthesis of pyrazole-based iron(III) complexes, exemplary for tris(1-phenylpyrazolato-$N,C^{2'}$)iron(III) (**Fe(ppz)₃**). (**Bottom**) Structures of tris(1-(4-(trifluoromethyl)phenyl)pyrazolato-$N,C^{2}$)iron(III) (**Fe(CF₃ppz)₃**), tris(1-(([1,1'-biphenyl])-4-yl)phenyl)pyrazolato-$N,C^{2}$)iron(III) (**Fe(bppz)₃**), tris(1-(naphthalen-2-yl)pyrazolato-$N,C^{2}$)iron(III) **Fe(naphpz)₃**, and tris(1-(4-methoxyphenyl)pyrazolato-$N,C^{2}$)iron(III) (**Fe(MeOppz)₃**), with their respective yields.

Single crystals suitable for X-ray diffraction were obtained by diffusion of cyclopentane into a solution of the respective complex in DCM. The exemplary crystal structure of **Fe(ppz)₃** is shown in Figure 2 (top), and crystallographic data for the other complexes are summarized in the Supplementary Information. All complexes show a distorted octahedral geometry with $C_3$ symmetry. While complexes with tris-bidentate ligands in general allow the formation of both facial (*fac*) and meridional (*mer*) isomers, only the formation of *fac*-**Fe(R-ppz)₃** is observed in all complexes' crystals [27,33,34]. This is most likely due to the trans effect, which leads to an enthalpically favored *fac*-isomer as in Co(ppz)₃ and other bidentate iridium, zinc, and iron complexes [27,35,36].

**Figure 2.** (**Top**) Single crystal structure of **Fe(ppz)₃**, displayed with 50% probability for the anisotropic displacement ellipsoids, hydrogen atoms omitted for clarity; (**Bottom**) *mer*-**Fe(ppz)₃** and *fac*-**Fe(ppz)₃** (with Δ- and Λ-enantiomers).

The *fac*-isomer allows for the formation of both Δ- and Λ-enantiomers, which were observed in direct adjacency within one unit cell. This was present in the structures of all complexes. Yet, it was not possible to isolate and quantify the enantiomer ratios.

Due to the racemic mixture, the three discrete phenylpyrazole-based ligands are crystallographically inequivalent, and the corresponding Fe-N and Fe-C bond lengths were therefore averaged. The key structural parameters are summarized in Table 1, which are in good agreement with the results obtained from DFT calculations (Table S17).

**Table 1.** Crystallographic data for the investigated iron(III) complexes; averaged over all binding distances.

| Complex | $d_{\varnothing}$ (Fe-N) (Å) | $d_{\varnothing}$ (Fe-C) (Å) | ∢Chelate Bite Angle (°) | ∢(C-Fe-N)$_{axial}$ (°) |
|---|---|---|---|---|
| **Fe(ppz)$_3$** | 2.0030(15) | 1.9508(13) | 87.07(7) | 171.54(7) |
| **Fe(CF$_3$ppz)$_3$** | 2.0075(15) | 1.9520(16) | 85.88(6) | 170.46(6) |
| **Fe(MeOppz)$_3$** | 2.0129(15) | 1.9512(13) | 94.52(5) | 170.89(5) |
| **Fe(bppz)$_3$** | 2.0122(7) | 1.9536(7) | 91.68(1) | 172.90(1) |
| **Fe(naphpz)$_3$** | 2.0134(12) | 1.9530(17) | 93.59(7) | 169.99(7) |

The Fe-C bonds of the presented complexes are about 0.05 Å shorter than the Fe-N bonds, probably due to the stronger donor properties plus π-accepting properties of the carbon. All bond lengths are approximately identical to those of Co(ppz)$_3$ [27]. Within the investigated set of compounds, the Fe-C bond lengths remain nearly identical; thus, the influence of the meta-substituents seems negligible. Nevertheless, the Fe-N bond length and chelate bite angles show slight variations in dependence of the different functional groups. The shortest Fe-N bonds and smallest bite angles are observed for **Fe(ppz)$_3$** (2.0030(15) Å, 87.07(7)°) and **Fe(CF$_3$ppz)$_3$** (2.0075(15) Å, 85.88°), while **Fe(MeOppz)$_3$** and **Fe(naphpz)$_3$** exhibit the longest Fe-N bonds (2.0129(15) Å and 2.0134(12) Å, respectively) and the largest bite angles (94.52(5)°, 93.59(7)°). Apparently, the resonance effect of the functional groups affects both Fe-N bond length and the chelate bite angle by altering electron density on the coordinating nitrogen. This would explain the similar bond lengths in **Fe(ppz)$_3$** and **Fe(CF$_3$ppz)$_3$**, since the trifluoromethyl group shows only minor resonance effects, whereas both the methoxy and naphthyl moieties show strong resonance effects. Therefore, the overall electron density on the ligand is increased, resulting in an elongated Fe-N bond and a higher chelate bite angle. In **Fe(bppz)$_3$** with a bite angle of 91.68(1)°, a rotation of the phenyl group to reduce the angular strain results in a weaker overlap of the π orbitals. Therefore, a less pronounced resonance effect is observed, resulting in shorter Fe-N bonds and smaller bite angles compared to **Fe(MeOppz)$_3$** and **Fe(naphpz)$_3$**. Influences due to steric effects are possible, but unlikely in this context. They may play a role in the C-Fe-N angle, which is largest in the unfunctionalized compound (171.54(7)°) and smallest in **Fe(naphpz)$_3$** (169.99(7)°), possibly due to repulsion of the ligands with the rigid naphthyl groups.

The presence of the +III oxidation state of iron could also be confirmed by NMR-spectroscopy (see Supplementary Information for further details), where relatively sharp resonances with a broad range of chemical shifts between 13.54 ppm and −75.20 ppm are observed. Assignment of $^1$H- and $^{13}$C-NMR signals could be achieved by various pulse sequences for paramagnetic compounds [37]. The chemical shifts of the complexes show an alternating pattern of the $^{13}$C resonances for both rings equally. An influence of the electron density distribution (Figure S56), as well as a resonance effect, cannot be excluded. Drastic proton resonance shifts of up to −75.20 ppm are observed for the ortho-positions of the cyclometalating phenylene, directly adjacent to the paramagnetic center. Interestingly, strong shielding of the protons adjacent to cyclometalating functions is also observed in diamagnetic complexes due to the proximity to the metal center and can be used as an indicator for successful cyclometallation [38–42]. The protons on the pyrazolyl-moiety are less affected, presumably due to the nitrogen atoms inhibiting the effects of the unpaired

electron originating from the iron. Additionally, the pyrazole, contrary to the phenylene, is not bound covalently to the iron center, which may also reduce the paramagnetic effects.

### 2.2. Cyclic Voltammetry and Optical Spectroscopy

The redox properties of the investigated complexes were investigated by cyclic voltammetry in MeCN (Figure 3). The measured solutions ($10^{-3}$ M) of the individual compounds contained tetrabutylammonium hexafluorophosphate ($[Bu_4N]PF_6$) in 0.1 M concentration as an electrolyte. The values reported in the following are referenced against $Fc/Fc^+$. The key results are summarized in Table 2.

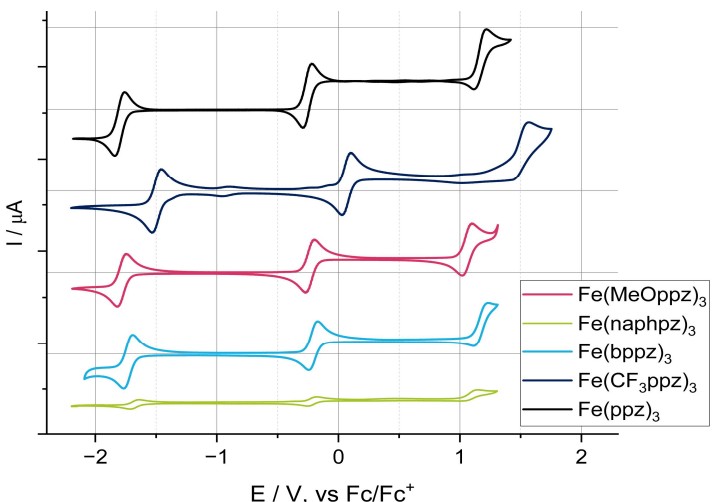

**Figure 3.** Cyclic voltammograms of the five investigated complexes ($10^{-3}$ M) in MeCN with 0.1 M $[Bu_4N]PF_6$ as electrolyte at a scan rate of 100 mV/s under light exclusion.

**Table 2.** Electrochemical [a] and electronic [b] and properties of the complexes.

| Complex | $E_{1/2}$ $Fe^{II/III}$ (V) | $E_{1/2}$ $Fe^{III/IV}$ (V) | $E_{1/2}$ (Ligand) (V) | $\Delta E_{LMCT}$ [e] (V) | $\lambda_{abs\text{-}max}$ (nm) ($\varepsilon$ ($cm^{-1}$ $M^{-1}$)) |
|---|---|---|---|---|---|
| **$Fe(ppz)_3$** | −1.80 (rev) | −0.26 (rev) | 1.17 (irrev) | 2.97 (418 nm) | 109 (6.18) 346 (1.63) 450 (0.64) ($\lambda_{max}$ = 522) [c] |
| **$Fe(CF_3ppz)_3$** | −1.50 (rev) | 0.07 (rev) | 1.54 [d] (irrev) | 3.04 (408 nm) | 293 (2.82) 350 (0.61) 417 (0.38) ($\lambda_{max}$ = 530) [c] |
| **$Fe(MeOppz)_3$** | −1.78 (rev) | −0.23 (rev) | 1.11 (irrev) | 2.89 (429 nm) | 290 (6.45) 356 (0.62) 453 (0.39) ($\lambda_{max}$ = 580) [c] |
| **$Fe(bppz)_3$** | −1.73 (rev) | −0.21 (rev) | 1.17 (irrev) | 2.90 (428 nm) | 277 (2.99) 343 (0.68) 440 (0.45) ($\lambda_{max}$ = 540) [c] |
| **$Fe(naphpz)_3$** | −1.68 (rev) | −0.22 (rev) | 1.08 (irrev) | 2.76 (449 m) | 284 (3.35) 362 (0.93) 442 (0.68) ($\lambda_{max}$ = 590) [c] |

[a] Concentration of $10^{-3}$ M in MeCN with 0.1 M $[Bu_4N]PF_6$ as electrolyte. [b] Molarity of $10^{-5}$ M in BuCN. [c] $\lambda_{max}$ = maximal value of absorption wavelength. [d] Anodic peak potential. [e] Calculated from the difference of $E_{1/2}$(ligand) and $E_{1/2}$($Fe^{II/III}$).

Three redox processes were identified in the potential window for all complexes. In the range of −0.26 to 0.07 V, transitions are observed that can be attributed to an iron(III/IV) redox process [18,28,43]. These values are similar to those found for the aforementioned $[Fe(ImP)_2]^{2+}$ (0.08 V), a compound containing two cyclometalating moieties [18]. Functionalization of the meta-position in the phenyl unit does not result in significant changes in the

iron(III/IV) redox potentials relative to **Fe(ppz)₃**, with the exception of **Fe(CF₃ppz)₃**. Here, an anodic shift of the oxidation potential is observed, in agreement with a stabilization of the metal-based levels by the electron-withdrawing CF₃ group. The same behavior is found for the iron(II/III) redox processes, which are observed between $-1.50$ and $-1.80$ V.

Above potentials of 1 V, irreversible ligand-based oxidation takes place. Although these processes are irreversible, the values can be used to discuss the influence of the different substituents on the electron density in the ppz ligand scaffold. While **Fe(bppz)₃** shows the same value as **Fe(ppz)₃**, the cathodically shifted oxidation potentials in **Fe(MeOppz)₃** and **Fe(naphpz)₃** indicate an increased electron density on the coordinating phenylene. This is consistent with the effects observed in the crystal structures, where increased electron density leads to increased Fe-N bond lengths. Consequently, **Fe(CF₃ppz)₃** shows the most anodically shifted ligand oxidation. From the difference in the ligand oxidation potential and the iron(II/III) transition, electrochemical bandwidths for an LMCT transition can be obtained. These band gaps reflect the same trends since all complexes exceed the band gap of $[Fe(ImP)_2]^{2+}$ (2.47 V) significantly. The highest $\Delta E$ values are observed for **Fe(ppz)₃** (2.97 V) and **Fe(CF₃ppz)₃** (3.04 V).

To confirm these results, UV-Vis spectra were recorded in butyronitrile (BuCN) due to the superior stability of the complexes in this solvent (vide infra) and are shown in Figure 4.

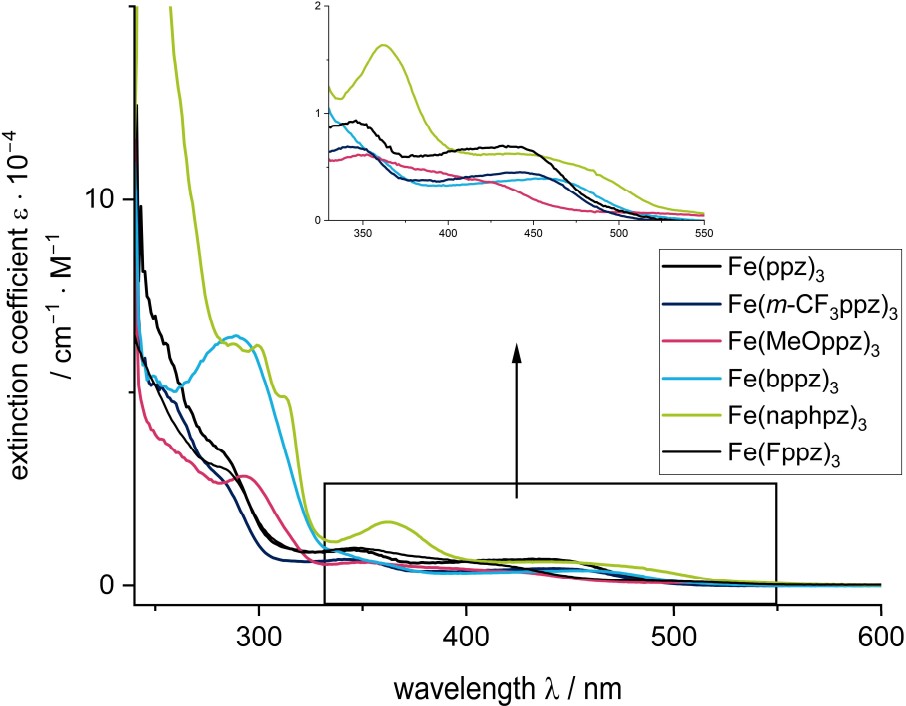

**Figure 4.** Absorption spectra in BuCN for the investigated compounds. Inset: Enhancement of the 330 to 550 nm absorption region.

The spectra can be divided into three regions. In the high-energy region below 325 nm, intense absorption bands can be observed. These are attributed to $\pi$–$\pi^*$ transitions. Consequently, **Fe(bppz)₃** and **Fe(naphpz)₃** show the highest intensities due to the extended $\pi$-systems.

Between 350 and 400 nm, the complexes exhibit a pronounced feature, which is the most intense for **Fe(naphpz)₃** at 362 nm. The remaining complexes show less intense, blue-shifted signals at 346 nm for **Fe(ppz)₃**, 350 nm for **Fe(CF₃ppz)₃**, 356 nm for **Fe(MeOppz)₃**, and 343 nm for **Fe(bppz)₃**. The origin of this absorption is presumably an MLCT [27].

Above 375 nm and 400 nm (for **Fe(naphpz)₃**), all complexes exhibit a broad absorption band. DFT calculations (vide infra, Figure 5) indicate that this band is composed of both MLCT and LMCT transitions, with a dominating MLCT character. This is unexpected, as

previously reported photoactive iron(III) complexes show an energetically lowest LMCT absorption and, in general, MLCT absorption only as an exception [18–20]. Therefore, the ligand design of this complex is responsible for this unexpected behavior. The interplay of pyrazole and phenylene as donors creates π and π* orbitals, which are in the right energetic distance to the metal orbitals to enable the energetically lowest mixed MLCT/LMCT bands with a dominating MLCT character. This may also be the reason that these compounds do not exhibit a fluorescence as could be expected for iron(III) in this strong donor environment. As stated in the literature, the excited $^2$MLCT states may undergo intersystem crossing (ISC) into a $^4$MLCT state, which can possibly relax into energetically lower $^4$MC states [22]. A possible explanation is that breathing and deformation modes in these bidentate complexes lead to this deactivation pathway, which is not possible in the more rigid tridentate [Fe(ImP)$_2$]$^+$, which shows emission from both $^2$MLCT and $^2$LMCT [18,44,45].

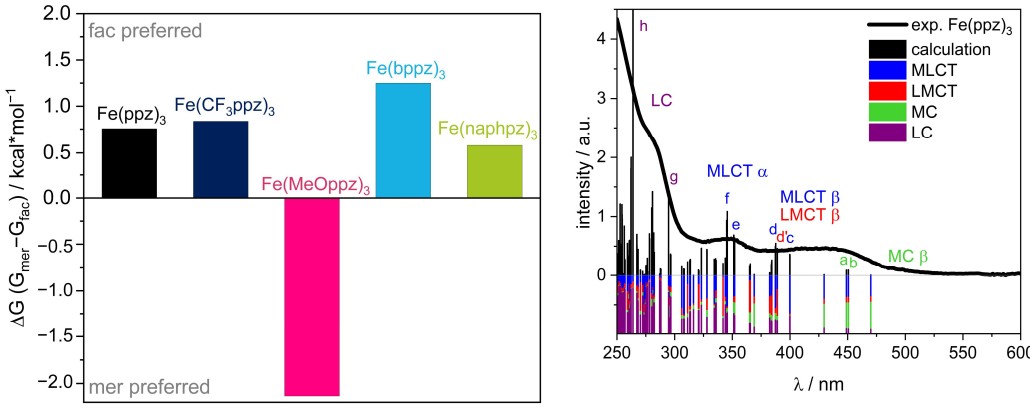

**Figure 5.** (**Left**) Gibbs energy difference (ΔG) between the *fac-* and *mer-*isomers of optimized geometries computed at the DFT/PBEh-3c level of theory. (**Right**) TPSSh-calculated vertical transitions for the *fac-*optimized structure of Fe(ppz)$_3$ in comparison with the experimental UV-Vis spectrum. Analysis of excited states is shown from 0 to −1. Further analysis of designated transitions a–h can be found in the Supporting Information (Table S18).

## 3. Computational Calculations

The formation of *fac-*isomers as a single product was confirmed by the crystal structure. This observation was further investigated by DFT calculations. For this purpose, the Gibbs free energy (G) for the optimized structures, using the PbEh-3c composite method, was calculated (Figure 5) [46]. For the complexes **Fe(ppz)$_3$, Fe(CF$_3$ppz)$_3$, Fe(bppz)$_3$**, and **Fe(naphpz)$_3$** the *fac-*isomer is energetically favored by 0.5–1.5 kcal·mol$^{-1}$. The higher stability is also reported for similar complexes such as Co(ppz)$_3$ and Ir(ppy)$_3$ and can be explained by the position of the phenyl groups, relative to the pyrazolyl groups, as elaborated on above [27,36]. Surprisingly, calculations suggest a higher thermodynamic stability for *mer-***Fe(MeOppz)$_3$** despite the fact that it crystallizes in the *fac-*isomer. This indicates a significant kinetic inhibition for the transformation of the kinetic product *fac-***Fe(MeOppz)$_3$** into the more stable *mer-***Fe(MeOppz)$_3$**. Furthermore, the calculated bond lengths and angles of the optimized *fac-*isomers match the crystal structure, indicating a good agreement between the calculation and the experimental values (Table S17). Only the angles of the oppositely lying C-Fe-N atoms are diverging between the experiment and the theoretical calculation, showing a better approximation of a perfect octahedral structure which is typical for a gas-phase optimized structure in comparison to the solid crystal structure.

To further explain the experimental findings, TD-DFT calculations with the meta-hybrid TPSSh functional and the def2-TZVP basis set were conducted [47,48]. The energies of the resulting frontier orbitals are shown in Figure 6. The spatial distribution can be found in the Supporting Information (Figure S56). Since all complexes are open-shell systems and thus the electron density of up (α) and down (β) spin is separated in the calculations,

two sets of singly occupied orbitals in the molecular orbital schemes are obtained. Due to the single Slater determinant in DFT as a reference function, such a multiconfigurational character is not well described, so the results must be considered with care.

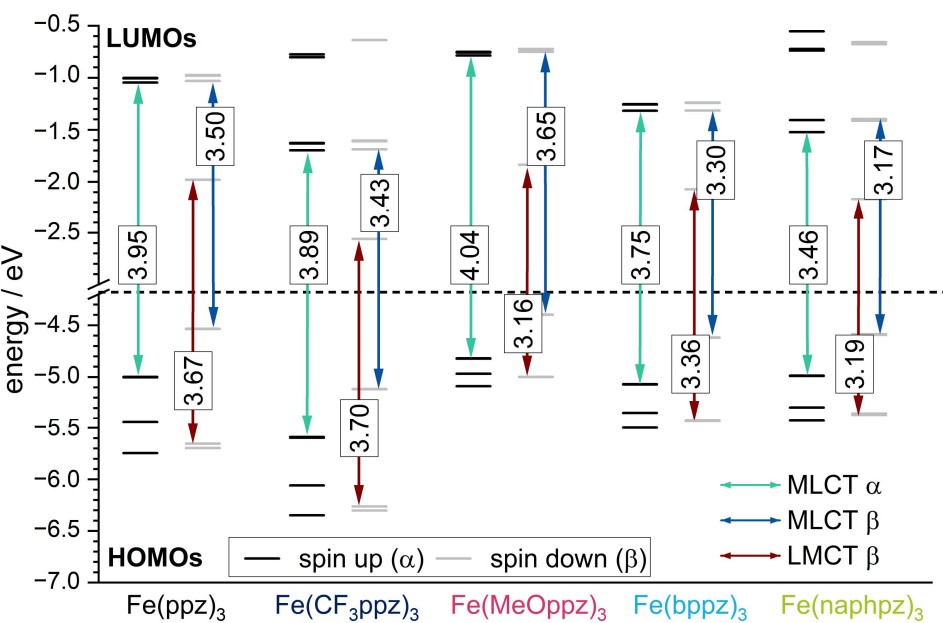

**Figure 6.** TPSSh-calculated molecular orbital schemes of all investigated complexes showing the HOMOs and LUMOs of the DFT-calculated spin-up and spin-down orbitals.

The β HOMO energy levels show mainly metal character (Figure S56) and can be correlated to the iron(III)/iron(IV) redox potentials. The trend of the energetic order of the calculated HOMOs follows the experimental CV data. In particular, **Fe(CF$_3$ppz)$_3$** shows the most stabilized HOMO at $-5.12$ eV, which is in line with the highest oxidation potential of 0.07 eV. For the remaining complexes, HOMO energy values with only small deviations of $-4.40$ eV, $-4.54$ eV, $-4.59$ eV, and $-4.62$ eV are obtained for **Fe(MeOppz)$_3$**, **Fe(ppz)$_3$**, **Fe(naphpz)$_3$**, and **Fe(bppz)$_3$**, respectively, which is in agreement with the small deviations for the experimental potentials of $-0.23$ eV, $-0.26$ eV, $-0.22$ eV, and $-0.21$ eV, respectively. The metal-based β LUMO can be correlated to the iron(II)/(III) redox potential. Since a lower negative potential for the reduction correlates with a lower orbital energy, **Fe(CF$_3$ppz)$_3$** shows the lowest negative potential of $-1.50$ eV and, as expected, has the lowest LUMO energy of $-2.56$ eV in the calculations. This is followed by slightly increasing calculated LUMO energies of **Fe(naphpz)$_3$** ($-2.17$ eV), **Fe(bppz)$_3$** ($-2.07$ eV), **Fe(MeOppz)$_3$** ($-1.84$ eV), and **Fe(ppz)$_3$** ($-1.98$ eV), reflecting the increasing negative potentials of $-1.68$ eV, $-1.73$ eV, $-1.78$ eV, and $-1.80$ eV, respectively. In addition, the lowest ground state (not including excited states) LMCT energies, which can be extracted from the energy differences of the HOMO−2 β and the LUMO β (Figure 6, red arrow), can be compared to the $\Delta E_{LMCT}$, calculated from CV potentials. Although the consideration of the difference of the frontier orbitals always yields slightly higher values of 0.3–0.7 eV than the experimental potential differences, the order for **Fe(CF$_3$ppz)$_3$**, which has the highest expected LMCT energy of 3.70 eV (calculated) and 3.04 eV (experimental), **Fe(ppz)$_3$** (3.67 eV/2.97 eV), **Fe(bppz)$_3$** (3.36 eV/2.90 eV), and **Fe(naphpz)$_3$** (3.19 eV/2.76 eV) remains the same. Only **Fe(MeOppz)$_3$** (3.16 eV/2.89 eV) shows the smallest energy difference in the calculations, while for the experimental $\Delta E_{LMCT}$, it shows the second smallest. To substantiate the assignment of the optical absorption bands, the first 150 vertical transitions (Figure S56) were calculated for the example of **Fe(ppz)$_3$** to determine the character of the bands in the UV-Vis spectrum. The lowest energy transitions in the visible area (a, b) originate from β HOMO to β LUMO+X transitions, with both orbitals having metal character, which is visualized in the Supporting Information (Table S18). Since the energy for the

vertical transitions is typically slightly overestimated, these transitions may be assigned to the shoulder at 500 nm in the experimental spectrum [49]. This is supported by the weak oscillator strength, characteristic of dipole-forbidden MC transitions. The broad band at about 425 nm can be assigned to both MLCT (c,d) and LMCT (d') transitions in the β-orbital set. This is consistent with almost equal energetic differences of the respective orbital levels shown in Figure 6, depicting the lowest-energy MLCT and LMCT states. Only for **Fe(MeOppz)₃** is the lowest LMCT energy below the MLCT energy, which explains the slightly changed experimental UV-Vis spectrum for this complex. The signal at 350 nm can be assigned to MLCT transitions (e, f) in the α-orbital set. For **Fe(naphpz)₃**, the red shift of the α MLCT band to 362 nm can be explained on the basis of the α HOMO-LUMO gap displaying the lowest-energy MLCT. Due to its large π-system leading to energetically low π* orbitals, the MLCT transition is significantly lower energetic in contrast to the remaining complexes. This is also reflected in the transitions below 300 nm, which can be assigned to ligand-centered (LC) transitions (g,h) and follow the general trend.

## 4. Hard X-ray Spectroscopy

The electronic structure of selected complexes was further investigated by synchrotron X-ray spectroscopy. X-ray absorption (XAS) and X-ray emission (XES) spectroscopy are useful methods to obtain structural and electronic information about metal complexes [50]. In the pre-edge of the X-ray absorption near edge structure (XANES) region, transitions from the 1s to the lowest unoccupied molecular orbital (LUMO) occur in K-edge spectra. For $d^5$ transition metals, the LUMO usually contains high fractions of the metal d orbitals. Since 1s → nd transitions are dipole-forbidden, intensity increases due to ligand-mediated hybridization with metal p-orbitals. Since the overlap depends on the geometry and symmetry of the complex, information about these factors is obtained [51]. The prepeak also provides information about the oxidation state by its energy. To enhance the experimental resolution, a specified emissive final state can be detected as the signal linewidth is inversely proportional to the lifetime of the measured final states (HERFD-XANES) [52,53]. XES examines the relaxation processes after the photoionization described above [54,55]. Core-to-core (CtC) XES spectra result from 3p → 1s transitions, where information about the spin state, the oxidation state of the metal, and the covalency of the bond between the ligand and metal is obtained due to the 3p–3d exchange interaction. 3d → 1s transitions are the origin of valence-to-core (VtC) XES spectra [56].

The XANES spectra of **Fe(ppz)₃**, **Fe(CF₃ppz)₃**, and **Fe(bppz)₃** shown in Figure 7a,b look nearly identical, indicating the very similar chemical and electronic structure in all three compounds. Due to the non-inversion symmetric character of the $C_3$-point group, hybridization leads to two prepeak signals at 7111 eV and 7114 eV. These two prepeaks imply accessible empty states in the non-degenerate ligand field states. TD-DFT calculations based on the TPSSh functional along with the def2-TZVPP basis set and the def2/J auxiliary basis set furthermore show that the first signal at 7111 eV can be ascribed to iron $d_{xz}$-orbitals as acceptor orbitals. The second signal at 7114 eV shows the transitions to the $d_{x2-y2/z2}$-orbital set. Due to the larger number of available holes in this d-orbital set, a much higher intensity can be observed here, indicating a low-spin state. This is also confirmed by CtC spectra (Figure S57), showing just a small splitting between the $Kb_{1,3}$ main line at around 7058 eV and the Kb' signal at 7045 eV for the three complexes. Furthermore, the signal in the XANES spectra does not shift for any of the complexes, confirming the same oxidation states. In the further course, two features appear at 7120 and 7124 eV. The first signal at 7120 eV emerges mainly from transitions into the pyrazole $C_p$-orbitals, and the signal at 7124 eV can be related to transitions into the phenyl $C_p$-orbitals. While the spectra at 7120 eV overlap exactly, small differences become apparent in the further course at 7124 eV. This can be attributed to the different substituents on the phenyl, shifting the $C_p$ orbitals in energy. The slight shift to higher energy follows the order **Fe(CF₃ppz)₃** < **Fe(ppz)₃** < **Fe(bppz)₃** and shows the electron-withdrawing and -donating effects of $CF_3^-$ and phenyl substituents, respectively.

The VtC spectra shown in Figure 7c,d show a similar behavior. For analysis, DFT calculations using the TPSS functional along with the def2-TZVPP basis set and def2/J auxiliary basis set for the RI-J approximation were used. The calculated spectra correspond well to the experimental data. This allows the main peak at 7109 eV to be attributed to transitions from the phenyl $C_p$-orbital with an admixture of $Fe_d$ orbitals (~20%) in all cases. The shoulder at 7106.5 eV can be attributed to transitions from the $N_p$ orbitals of the pyrazole with an admixture of $Fe_d$ orbitals. At lower energies, the cross-over transitions are located. From 7101 to 7105 eV, there are transitions from the phenyl and pyrazole p-orbitals, and around 7085 to 7100 eV, there are transitions from ligand s-orbitals.

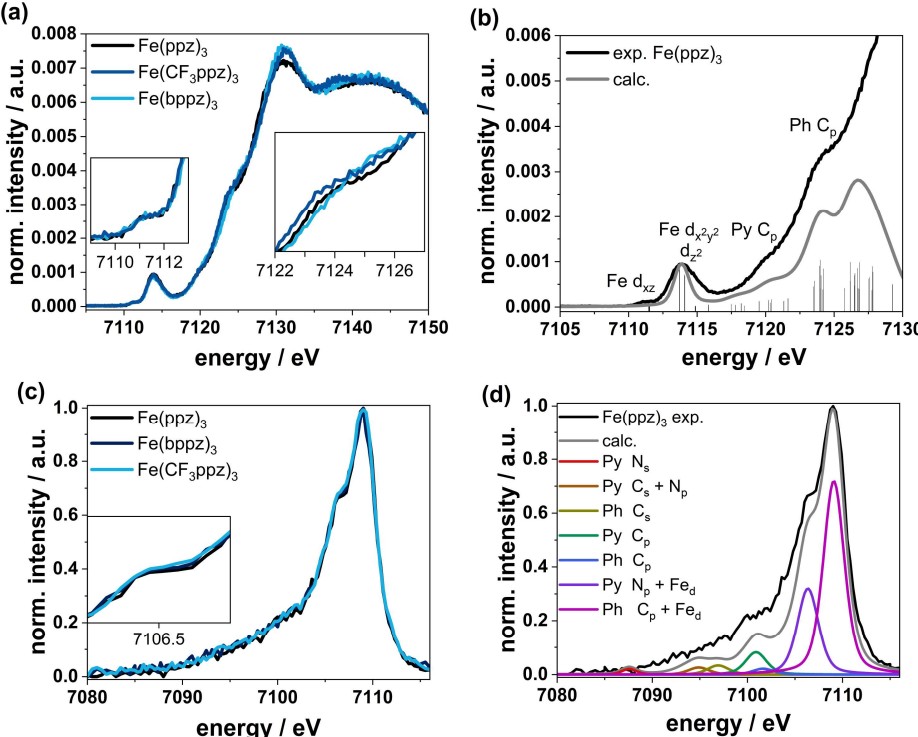

**Figure 7.** Experimental XANES (**a**) and VtC (**c**) spectra of Fe(ppz)₃, Fe(bppz)₃, and Fe(CF₃ppz)₃) and comparison of experimental and calculated XANES (**b**) and VtC (**d**) spectra exemplary for Fe(ppz)₃ with main character of acceptor (**b**) and donor (**d**) orbitals accountable for the peak (calculated transitions for Fe(bppz)₃ and Fe(CF₃ppz)₃ can be found in the Supporting Information Figure S58).

## 5. Behavior under Irradiation

To explore potential applications of the investigated complexes as photoactive compounds, their behavior under continuous irradiation was explored. Photodecomposition of all complexes is observed in acetonitrile under broadband irradiation (300 W xenon lamp, 390–800 nm) for 24 h (Figure S51). ¹H-NMR measurements of the resulting products showed signals attributed to a ligand-based product and an NMR-silent iron species. With assistance from mass spectroscopy (found: $m/z_{exp}$ = 287.1268) and additional 2D-NMR spectra, the C-C-homocoupled ligand–dimer 2,2'-di(1*H*-pyrazol-1-yl)-1,1'-biphenyl ($m/z_{theo}$ = 286.1218) was identified as the main decomposition product after illumination of **Fe(ppz)₃**. This behavior and the respective product of the reductive elimination can be observed for all complexes described here.

Accordingly, irradiated **Fe(ppz)₃** and the functionalized complexes undergo reductive ligand elimination. The C-C-coupling mechanism (Figure 8) is presumably close to the related Co(ppz)₃ complex, which was already investigated by Thompson et al. [27].

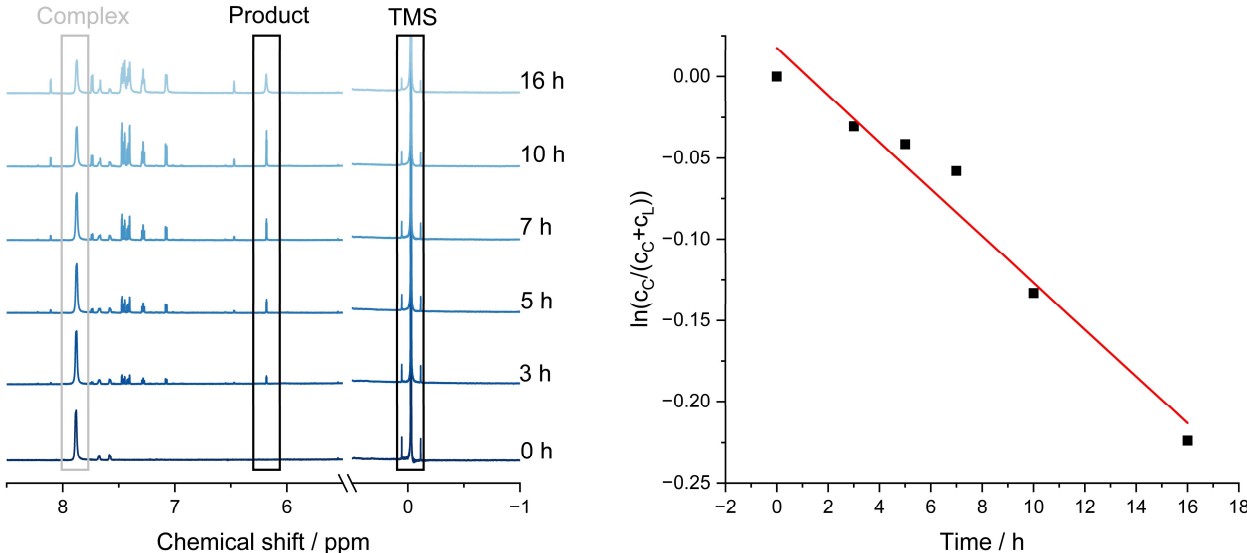

**Figure 8.** Proposed separation mechanism after illumination of the initial **Fe(ppz)₃** complex towards 2,2′-di(1H-pyrazol-1-yl)1,1′biphenyl. Adapted from [27].

To determine the rate constant of the ligand-separation process, time-dependent ¹H-NMR measurements were carried out, and the dynamic signals were referenced against TMS, exemplarily conducted for **Fe(ppz)₃**.

Within the first 5 h of irradiation, approximately 4% of the complex disintegrated; after 16 h, it was 15%. This means at least 85% of the complex is still intact even after 24 h under irradiation, following a first-order reaction (Figure 9, right) and a decomposition constant of $k = -0.0144 \text{ h}^{-1}$.

**Figure 9.** (**Left**) NMR spectra after irradiation for (from bottom to top) 0, 3, 5, 10, and 16 h, referenced against TMS; (**Right**) decomposition of **Fe(ppz)₃**, time against relative concentration of the signal at 6.21 ppm for two protons at the coupled ligand and 7.90 ppm, for three protons at the complex. Calculation included the relative intensity, multiplied by the fracture of the protons. The linear regression shows a slope of $-0.0144 \text{ h}^{-1}$ [57].

Additionally, the stability of **Fe(ppz)₃** under irradiation was also investigated in different solvents by ¹H-NMR spectroscopy. Measurements in acetone, DMSO, THF, dichloromethane, methanol, benzene, toluene, chloroform, butyronitrile (BuCN), and 2-methyltetrahydrofuran (2-MeTHF) (Figure S52) showed that the complex was only broadband-stable in the last two solvents. The reason for this behavior needs to be explored in the future. The UV-Vis spectra had to be accordingly recorded in BuCN, and the CV was conducted under light exclusion.

Finally, monitoring of the decomposition of **Fe(ppz)₃** at various wavelengths was analyzed (Figure S53). By the installation of bandwidth filters at 320, 360, and 390 nm, as

well as long-pass filters at 400 and 495 nm, wavelength-dependent measurements were enabled. Decomposition only occurred for wavelengths higher than 400 nm and lower than 495 nm. Therefore, excitation into the broad band composed of β $^2$LMCT and β $^2$MLCT transitions leads inevitably to the decomposition of the compound, whereas the $^2$MC transitions do not lead to decomposition. A tentative explanation reiterates the explanation for the absence of luminescence: Excitation of the $^2$MLCT state leads to ISC into the $^4$MLCT state, which relaxes into a $^4$MC state. Due to the population of the antibonding metal-based orbitals, the bonds of the pyrazole to the iron are elongated further or completely dissociated, as seen in photochemically induced ligand release, allowing the coordination of solvent molecules and possibly leading to reductive elimination [58]. This is supported by the optimized structure of the lowest quartet state. The spin density plot of this state (Figure S55) shows that it has a metal-centered character. In addition, a significant bond expansion of one Fe-N bond from 2.0 Å in the $^2$GS to 2.4 Å in the $^4$MC state occurs, which might indicate the first step of the decomposition process.

## 6. Conclusions

The synthesis of five different homoleptic iron(III) complexes with bidentate phenylpyrazole-based ligands was reported in this work, with functional groups on the 4-position of the phenyl ring, such as trifluoromethyl, methoxy, phenyl, and naphthyl groups, to compare the influence of electron-donating or -withdrawing groups on the electronic structure. All compounds were received purely as *fac*-isomers in low-spin iron(III) configuration. Despite the paramagnetism, the complete NMR signal assignments and substantiated DFT calculations could be performed.

The complexes show very low-lying oxidation potentials of −0.26 V for the parent **Fe(ppz)₃**, where the strong σ-donating capabilities of the phenyl moiety impact the redox behavior drastically compared to common polypyridyl complexes. The effect of electron-withdrawing moieties consequently shifts the potentials to 0.07 V in **Fe(CF₃ppz)₃**, whereas the reduction potentials are more affected by electron-donating groups.

The absorption behavior, as assigned by TDDFT, is dominated by LC transitions in the UV range, whereas two absorption bands are observed above 350 nm. The higher energy band around 350 nm can be attributed solely to α MLCT transitions, and the broad and featureless absorption band above 370 nm (400 nm for **Fe(naphpz)₃**) is assigned to a mixture of MLCT and LMCT transitions, with an MC transition as the lowest energy shoulder. This unexpected behavior is caused by the ligand design, incorporating both cyclometalating and pyrazole ligands, leading to isoenergetic π-E* and E-π* gaps, which finally causes overlapping LMCT and MLCT transitions.

No emission of the complexes is observed. Instead, decomposition by reductive elimination is caused by irradiation over a longer period in the energy range of 400–495 nm. Therefore, an excited state relaxation following a $^2$MLCT → $^4$MLCT → $^4$MC cascade, leading to the population of a non-emissive and destructive $^4$MC state, is most likely. There, elongated or dissociated bonds allow for the coordination of solvent molecules and the reductive elimination of two homocoupled ligands.

With the presented results, the first spectroscopic and theoretical characterization of tris-bidentate iron(III) complexes is provided. The introduction of further functional groups in different positions or exchanging the pyrazole for other donor groups would potentially suppress reductive elimination, and photostable and photoactive compounds may be obtained.

## 7. General Procedures

*Complex Synthesis*

The described synthetic procedure applies to all complexes.

Ligand (3 equiv) was suspended in tetrahydrofuran (THF) (10 mL) under an argon atmosphere. Ethylmagnesium bromide (4 equiv, 0.9 M in THF) was added dropwise and refluxed overnight. In a second flask, iron powder (12 equiv) was added to a THF solution

of iron(II) bromide (1.5 equiv) (40 mL) and refluxed overnight. After refluxing, the flask was cooled to room temperature, and the ligand solution was cooled in an ethanol–nitrogen bath to −80°C. The iron(II) bromide solution was added dropwise and slowly warmed to room temperature. To the reaction mixture, a solution of $NH_4Cl$ (100 mL, 15 g/L) was added and extracted with dichloromethane (DCM) (3 × 100 mL). The combined organic phases were dried with $MgSO_4$ and concentrated under reduced pressure. Column chromatography with silica as solid phase and DCM as eluent was applied. The combined fractions were concentrated under reduced pressure and crystallized with slow diffusion of cyclopentane into the DCM-analyte solution. After removing the crystalline product and drying it at 50 °C under vacuum, the compound was received as an elemental-analysis pure product.

All experimental data can be found in the Supporting Information.

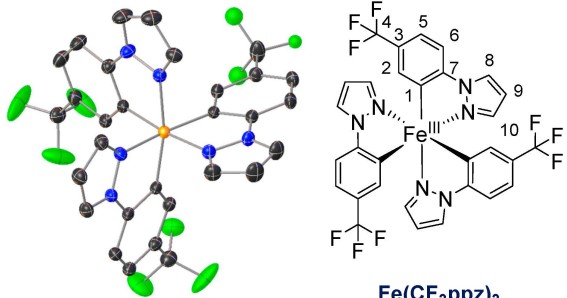

**Fe(ppz)$_3$**

### Tris(1-phenylpyrazolato-*N*,*C²′*)iron(III)Fe(ppz)$_3$

The complex was obtained as a yellow powder (7.5%).

**$^1$H-NMR** (700.0 MHz, CD$_3$CN): δ = −75.20 (s, 1H, 2-*H*), −9.27 (s, 1H, 9-*H*), −5.25 (s, 1H, 4-*H*), −3.35 (s, 1H, 8-*H*), −1.15 (s, 1H, 5-*H*), 7.90 (s, 1H, 3-*H*), 13.04 (s, 1H, 7-*H*) ppm. **$^{13}$C-NMR** (176.1 MHz, CD$_3$CN): δ = −92.7 (1C, 3-*C*), −74.8 (1C, 5-*C*), 100.2 (1C, 7-*C*), 117.0 (1C, 8-*C*), 122.9 (1C, 9-*C*), 239.9 (1C, 4-*C*), 389.1 (1C, 2-*C*), 396.6 (1C, 6-*C*) ppm. **$^{15}$N-NMR** (70.96 MHz, CD$_3$CN): 70.4 ppm. (ESI in MECN): *m/z* 485.1160 (for C$_{27}$H$_{21}$FeN$_6$ calc. 485.1177) **Elemental analysis**: calc. for C$_{27}$H$_{21}$FeN$_6$ C: 66.82%, H: 4.36%, N: 17.32%, found: C: 66.77%, H: 4.56%, N: 17.25%. **IR** (ATR, $\widetilde{v}$ [cm$^{-1}$]): 3139 w, 3041 w, 1573 w, 1506 w, 1461 m, 1434 m, 1417 m, 1398 m, 1328 w, 1270 m, 1236 w, 1193 w, 1153 w, 1099 w, 1064 m, 1043 m, 1012 m, 960 m, 918 w, 871 w, 825 w, 742s, 715 m, 698 m, 661 w, 644 w, 609 m.

**Fe(CF$_3$ppz)$_3$**

### Tris(1-(4-(trifluoromethyl)phenyl)pyrazolato-*N*,*C²*)iron(III)Fe(CF$_3$ppz)$_3$

The complex was obtained as a yellow powder (17.2%).

**$^1$H-NMR** (700.0 MHz, CD$_3$CN): δ = −75.17 (s, 1H, 2-*H*), −10.47 (s, 1H, 10-*H*), −3.39 (s, 1H, 9-*H*), −3.17 (s, 1H, 5-*H*), 0.27 (s, 1H, 6-*H*), 11.23 (s, 1H, 8-*H*) ppm. **MS** (ESI in MECN): *m/z* 689.0800 (for C$_{30}$H$_{18}$F$_9$FeN$_6$ calc. 689.0799). **$^{13}$C-NMR** (176.1 MHz, CD$_3$CN): δ = −93.4 (1C, 3-*C*), −68.6 (1C, 6-*C*), 108.9 (1C, 8-*C*), 112.6 (1C, 9-*C*), 126.5 (1C, dd, $^1J_{CF}$ = 272.85 Hz 275.09 Hz, 4-*C*), 132.8 (1C, 10-*C*), 229.9 (1C, 5-*C*), 361.9 (1C, 2-*C*), 382.0 (1C, 7-*C*) ppm. **$^{15}$N-NMR** (70.96 MHz, CD$_3$CN): 84.0 ppm. **$^{19}$F-NMR** (659.0 MHz, CD$_3$CN): δ = −71.5 (s, 3F) ppm. **Elemental analysis**: calc. for C$_{30}$H$_{18}$F$_9$FeN$_6$: C: 52.27%, H: 2.63%, N: 12.19%, found: C: 51.90%, H: 2.81%, N: 12.12%. **IR** (ATR, $\widetilde{v}$ [cm$^{-1}$]): 3155 w, 3033 w, 2360 w, 2335 w, 1585 w, 1508 w, 1477 w, 1396 m, 1315s, 1272s, 1249 m, 1159 m, 1110s, 1066s, 1045s, 960 m, 900 m, 838 w, 821 w, 804 m, 746s, 702 m, 661 m, 607 w.

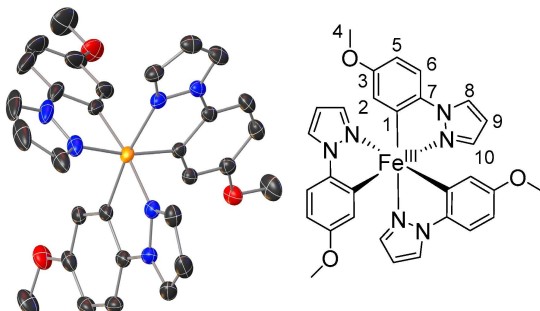

**Fe(OMeppz)₃**

**Tris(1-(4-methoxyphenyl)pyrazolato-*N*,*C²*)iron(III) Fe(MeOppz)₃**
The complex was obtained as a red powder (2.6%).
**¹H-NMR (700 MHz, CD₃CN)**: δ = −79.93 (s, 1H, 2-*H*), −10.34 (s, 1H, 10-*H*), −5.36 (s, 1H, 6-*H*)), −5.12 (s, 1H, 5-*H*), −3.19 (s, 1H, 9-*H*), 1.44 (s, 3H, 4-*H*), 12.06 (s, 1H, 8-*H*) ppm.
**¹³C-NMR\*** (176.1 MHz, CD₃CN): δ = −106.1 (1C, 156.6 Hz, 6-*C*), −93.0 (1C, 3-*C*), 49.1 (1C, 139.07 Hz, 4-*C*), 103.9 (1C, 182.9 Hz, 8-*C*), 114.1 (1C, 191.7 Hz, 9-*C*) 130.1 (1C, 182.9 Hz, 10-*C*), 232.4 (1C, 159.9 Hz, 5-*C*), 360.8 (1C, 129.6 Hz, 2-*C*), 403.2 (1C, 7-*C*) ppm. **¹⁵N-NMR** (70.96 MHz, CD₃CN): 80.2 ppm. **MS** (ESI in MECN): *m/z* 575.1530 (for C₃₀H₂₇FeN₆O₃ calc. 575.1494).
**Elemental analysis**: calc. for C₃₀H₂₇FeN₆O₃: C: 62.62%, H: 4.73%, N: 14.61%, found: C:62.49%, H: 5.19%, N: 14.35%. **IR** (ATR, $\tilde{v}$ [cm⁻¹]): 3122 w, 3039 w, 3006 w, 2952 w, 2931 w, 2902 w, 2829 w, 1583 w, 1560s, 1506 w, 1469s, 1417s, 1315 m, 1276s, 1249 m, 1209s, 1174s, 1116 m, 1031s, 958 m, 879 m, 811 w, 784s, 744s, 661 w, 621 m, 609 m.
\* not decoupled.

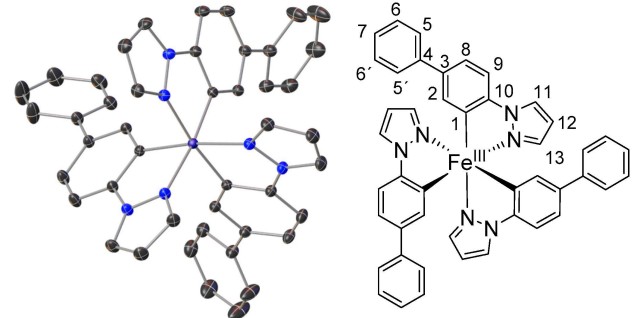

**Fe(bppz)₃**

**Tris(1-(([1,1′-biphenyl])-4-yl)phenyl)pyrazolato-*N*,*C²*)iron(III) Fe(bppz)₃**
The complex was obtained as a red powder (3.9%).
**¹H-NMR** (700 MHz, CD₃CN): δ = −77.61 (1s, 1H, 2-*H*) −10.18 (s, 1H, 13-*H*), −4.64 (s, 1H, 8-*H*), −3.21 (s, 1H, 12-*H*), −1.74 (s, 1H, 9-*H*), 5.39 (d, ³*J*HH = 8.19 Hz, 2H, 5,5′-*H*), 5.75 (t, ³*J*HH = 7.60 Hz, 2H, 6,6′-*H*), 7.10 (t, ³*J*HH = 7.14 Hz, 1H, 6-H) 11.84 (s, 1H, 11-*H*) ppm. **¹³C-NMR** (176.1 MHz, CD₃CN): δ = −79.6 (1C, 152.5 Hz, 9-*C*), −77.9 (1C, 3-*C*), 105.4 (1C, 185.7 Hz, 11-*C*), 114.4 (1C, 62.3 Hz, 12-*C*), 115.3 (2C, 89.03 Hz, 5,5′-*C*), 123.1 (1C, 162.9 Hz, 7-*C*), 130.9 (2C, 160.4 Hz, 6,6′-*C*), 131.9 (1C, 183.3 Hz, 13-*C*), 154.1 (1C, 4-*C*), 241.4 (1C, 158.8 Hz, 8-*C*), 370.5 (1C, 133.9 Hz, 2-*C*), 396.7 (1C, 10-*C*) ppm. **¹⁵N-NMR** (70.96 MHz, CD₃CN): 71.0 ppm. **MS** (ESI in MECN): *m/z* 713.2150 (for C₄₅H₃₃FeN₆ calc. 713.6460). **Elemental analysis:** calc. for C₄₅H₃₃FeN₆ C: 75.74%, H: 4.66%, N: 11.78%, found: C: 71.15%, H: 4.84%, N: 10.78%. **IR** (ATR, $\tilde{v}$ [cm⁻¹]): 3108 w, 3056 w, 3020 w, 1598 w, 1564 w, 1502 w, 1465 m, 1400 m, 1373 w, 1330 w, 1261 m, 1112 w, 1064 m, 1047 m, 1016 w, 958 w, 916 w, 894 w, 808s, 757s, 734s, 694s, 663 w, 649 w, 607 w.

**Fe(naphpz)₃**

**Tris(1-(naphthalen-2-yl)pyrazolato-*N,C²*)iron(III) Fe(naphpz)₃**
The complex was obtained as a red powder (5.3%).
**¹H-NMR** (700 MHz, DMSO-d6): δ = −85.19 (s, 1H, 2-*H*), −12.51 (s, 1H, 13-*H*), −1.68 (d, $^3J_{HH}$ = 6.3 Hz, 1H, 4-*H*), −1.31 (s, 1H, 12-*H*), 1.10 (s, 1H, 9-*H*), 1.25 (t, $^3J_{HH}$ = 7.10 Hz, 1H, 6-*H*), 8.12 (t, $^3J_{HH}$ = 6.30 Hz, 1H, 5-*H*), 11.15 (s, 1H, 11-*H*), 12.83 (d, $^3J_{HH}$ = 8.50 Hz, 1H, 7-*H*) ppm. **¹³C-NMR\*** (176.1 MHz, DMSO-d6): δ = −97.6 (1C, 3-*C*), −56.79 (1C, 150.66 Hz, 9-*C*), 80.3 (1C, 150.66 Hz, 7-*C*), 81.6 (1C, 160.34 Hz, 5-*C*), 99.4 (1C, 180.91 Hz, 12-*C*), 124.3 (2C, 188.74 Hz, 11-*C*), 126.6 (1C, 188.74 Hz, 13-*C*), 172.1 (2C, 157.98 Hz, 6-*C*), 180.5 (1C, 157.98 Hz, 4-*C*), 216.3 (1C, 8-*C*), 365.3 (1C, 2-*C*), 413.9 (1C, 10-*C*) ppm. **MS** (ESI in MECN): $m/z$ 635.1640 (for $C_{39}H_{27}FeN_6$ calc. 635.1647). **Elemental analysis**: calc. for $C_{39}H_{27}FeN_6$: C: 73.71%, H: 4.28%, N: 13.22%, found: C: 74.29%, H: 5.10%, N: 12.42%. **IR** (ATR, $\tilde{v}$ [cm$^{-1}$]): 3126 w, 6047 w, 2917 w, 2854 w, 1585 w, 1593 w, 1560 w, 1510 w, 1486 w, 1459 m, 1405 m, 1332 w, 1313 w, 1251 w, 1197 w, 1134 w, 1107 w, 1062 m, 1037 w, 977 w, 935 w, 889 w, 856 m, 831 w, 736s, 682 w, 651 w.
\* not decoupled.

## 8. Materials and Methods

Synthesis of ligands and complexes was carried out under standard Schlenk conditions, under inert and anhydrous conditions. Inert and pre-dried argon was used, and all applied glassware was heated under vacuum und flushed with inert gas three times. Anhydrous solvents were provided by a solvent drying plant from MBraun (MB SPS 800, München, Germany) and purged with argon prior to use. Used chemicals for all synthesis were commercially purchased from *Fischer Scientific* (Hampton, NH, USA), *Merck* (Rahway, NJ, USA), *Abcr* (Karlsruhe, Germany), and *TCI* (Chennai, India) and used without further purification. Ligand synthesis has been reported in [29,30].

### 8.1. NMR Spectroscopy

NMR spectra were recorded on a BRUKER Avance 700 (¹H, 700.1 MHz, Billerica, MA, USA) using deuterated solvents from *Deutero* (Kastellaun, Germany) without further purification. NMR signals were referenced to residual solvent signals relative to TMS. For the decomposition experiments a 300 W xenon lamp for irradiation was installed in front of an NMR sample of **Fe(ppz)₃**, exemplarily. To determine the percentage of intact complex, it was referenced against a calibrated TMS signal with the product species signal at 6.21 ppm, with common analytic calculations. Mass spectrometry was performed with a quadrupole time-of-flight mass spectrometer (MS) Synapt 2G from the company WATERS (Milford, MA, USA). Elemental analysis measurements were performed with a Micro Cube from ELEMENTAR (Langenselbold, Germany) and were compared with the theoretically calculated mass. A **PerkinElmer** (Waltham, MA, USA) Lambda 465 single-beam spectrophotometer was used for UV-Vis spectra. Solutions had a concentration of 10$^{-5}$ M BuCN and were measured in a **Hellma** (Müllheim, Germany) quartz cuvette with a path length of 1 cm. For IR spectroscopy a Bruker Vertex 70, with the sample as solid powder and the ATR technique, was applied. Cyclic voltammograms were measured with a 10$^{-3}$ M analyte and 0.1 M [*n*-Bu₄N](PF₆) concentration on a PGSTAT 101 potentiostat

from **Metrohm-Autolab** (Utrecht, The Netherlands). Emission spectra were recorded on an FLS1000 from **Edinburgh Instruments** (Livingston, UK) at room temperature.

### 8.2. Quantum Chemical Calculations

All calculations presented here were conducted with the ORCA quantum chemistry package (version 5.0.3) [59]. Unconstrained geometry optimization was performed using the PBEh-3c composite method [46], whereas a frequency calculation was performed and checked for the absence of negative values to confirm a minimum structure. Optimized structures were used as input for further calculations. Time-dependent (TD) DFT calculations for the extraction of orbital energies and the prediction of vertical transitions were conducted using the TPSSh functional [60] together with the def2-TZVP basis set, as well as the def2/J auxiliary basis set [61] and the RIJCOSX approximation [62] for the Hartree-Fock component. The tight convergence criterion was imposed on all calculations, and the D4 dispersion correction [62] was always employed when not using the PBEh-3c method. The conductor-like polarizable continuum model (CPCM) [63] for acetonitrile accounting for solvent effects was applied. For the simulation of the XANES and VtC XES spectra, the same settings except the CPCM model were used. For XANES, the TD-DFT approach with the TPSSh basis set and the expanded CP(PPP) basis set [64] only for iron was applied. VtC-XES spectra were calculated based on the DFT approach using the TPSS and CP(PPP) functional [60]. XANES transitions were plotted with linearly increasing broadening to higher energies, starting from 0.6 (fwhm) at the prepeak, and were shifted by 155.3 eV to match the experimental spectrum. VtC-XES transitions were broadened by 2.5 eV (fwhm), and all spectra were shifted by 170.6 eV. Ligand or atom projected VtC-XES spectra were created by taking only a set of donor orbitals with significant populations of a given atom or fragment into account. The analysis of the fractions of the molecular orbitals was based on the Löwdin population analysis, which was extracted from the ORCA output file using MOAnalyzer (version 1.3) and the TheoDORE package [65,66]. Spatial distributions of orbitals were visualized using IboView (version 20150427) [67].

### 8.3. X-ray Absorption and Emission Spectroscopy

X-ray absorption and emission experiments were performed at beamline ID26 at the European Synchrotron Radiation Facility (ESRF) in Grenoble [68]. The electron energy was 6.0 GeV, and the ring current varied between 180 and 200 mA. Incident energy calibration was performed using a Fe foil. For K-edge measurements, the solid samples were prepared as wafers using degassed cellulose as a binder to avoid self-absorption effects. The XANES spectra were monitored using a photodiode installed at about a 90° scattering angle and at 45° to the sample surface. To exclude radiation damage, fast measurements over the prepeak were carried out under the measurement conditions (attenuated beam, cryostat to cool the sample to 80 K). No signs of radiation damage could be detected. VtC-XES spectra were recorded at an excitation energy of 7300 eV measured with a Johann-type spectrometer [69].

### 8.4. Single-Crystal X-ray Diffraction

The presented X-ray single-crystal data were collected on a *Bruker Venture D8* three-cycle diffractometer equipped with a Mo $K_\alpha$ μ-source (λ = 0.71073 Å). Monochromatization of the radiation was obtained using *Incoatec* (Geesthacht, Germany) multilayer Montel optics, and a Photon III area detector was used for data acquisition. All crystals were kept at 120 K during measurement.

Data processing was carried out using the *Bruker* APEX 4 software package: This includes SAINT for data integration and SADABS for multi-scan absorption correction. Structure solution was obtained by direct methods, and the refinement of the structures using the full-matrix least squares method based on $F^2$ was achieved in SHELX [70,71]. All non-hydrogen atoms were refined anisotropically, and the hydrogen atom positions were refined at idealized positions riding on the carbon atoms with isotropic displacement

parameters $U_{iso}(H) = 1.2\ U_{eq}(C)$ and $1.5\ U_{eq}(-CH_3)$ and C-H bond lengths of 0.93–0.96 Å. All $CH_3$ hydrogen atoms were allowed to rotate but not to tip.

Crystallographic data were deposited at the Cambridge Crystallographic Data Centre and assigned the deposition numbers 2191100-2191104. Copies are available free of charge via www.ccdc.cam.ac.uk, accessed on 1 January 2020.

**Supplementary Materials:** The following supporting information can be downloaded at: https://www.mdpi.com/article/10.3390/inorganics11070282/s1, Figure S1: $^{1}$H-NMR spectrum of complex Fe(ppz)$_3$ in CD$_3$CN; Figure S2: $^{13}$C-NMR spectrum of complex Fe(ppz)$_3$ in CD$_3$CN; Figure S3: $^{15}$N-HMBC spectrum of Fe(ppz)$_3$ in CD$_3$CN, second signal is the folded signal of non-deuterated solvent CH$_3$CN; Figure S4: ESI-MS spectrum of complex Fe(ppz)$_3$ in CH$_3$CN; Figure S5: Complete cyclovoltammetry spectra for complex Fe(ppz)$_3$ in CH$_3$CN; Figure S6: Plotted data of Randles-Sevcik-Equation Fe(ppz)$_3$ at different scan rates, first redox step; Figure S7: Plotted data of Randles-Sevcik-Equation Fe(ppz)$_3$ at different scan rates, second redox step; Figure S8: Plotted data of Randles-Sevcik-Equation Fe(ppz)$_3$ at different scan rates, third redox step; Figure S9: ATR-IR spectrum for complex Fe(ppz)$_3$; Figure S10: Change in the absorptive behaviour of Fe(ppz)$_3$ with an applied potential of 0.5–2 V in CH$_3$CN; Figure S11: Change in the absorptive behaviour of Fe(ppz)$_3$ with an applied potential of $-0.5$ V in CH$_3$CN; Figure S12: Change in the absorptive behaviour of Fe(ppz)$_3$ with an applied potential of $-2.0$–2.5 V in CH$_3$CN; Figure S13: $^{1}$H-NMR spectrum of complex Fe(bppz)$_3$ in CD$_3$CN; Figure S14: $^{13}$C-NMR spectrum of complex Fe(bppz)$_3$ in CD$_3$CN; Figure S15: $^{15}$N-HMBC spectrum of Fe(bppz)$_3$ in CD$_3$CN; Figure S16: ESI-MS spectrum of complex Fe(bppz)$_3$ in CH$_3$CN; Figure S17: Cyclovoltammetry spectra of Fe(bppz)$_3$ in CH$_3$CN; Figure S18: Plotted data of Randles-Sevcik-Equation Fe(bppz)$_3$ at different scan rates, first redox step; Figure S19: Plotted data of Randles-Sevcik-Equation Fe(bppz)$_3$ at different scan rates, second redox step; Figure S20: Plotted data of Randles-Sevcik-Equation Fe(bppz)$_3$ at different scan rates, third redox step; Figure S21: ATR-IR- spectrum of complex Fe(bppz)$_3$; Figure S22:$^{1}$H-NMR spectra of complex Fe(CF$_3$ppz)$_3$ in CD$_3$CN; Figure S23: $^{13}$C-NMR spectra of complex Fe(CF$_3$ppz)$_3$ in CD$_3$CN; Figure S24: $^{15}$N-HMBC spectrum of the complex Fe(CF$_3$ppz)$_3$ in CD$_3$CN; Figure S25: $^{19}$F-HMBC spectrum of the complex Fe(CF$_3$ppz)$_3$ in CD$_3$CN; Figure S26: ESI-MS of complex Fe(CF$_3$ppz)$_3$ in CH$_3$CN; Figure S27: Cyclovoltammetry spectra of Fe(CF$_3$ppz)$_3$ in CH$_3$CN; Figure S28: Plotted data of Randles-Sevcik-Equation Fe(CF$_3$ppz)$_3$ at different scan rates, first redox step; Figure 29: Plotted data of Randles-Sevcik-Equation Fe(CF$_3$ppz)$_3$ at different scan rates, first redox step; Figure S30: ATR-IR-spectrum of complex Fe(CF$_3$ppz)$_3$; Figure S31: $^{1}$H-NMR spectrum of complex Fe(naphpz)$_3$ in DMSO-d$^6$; Figure S32: $^{13}$C-NMR spectrum of complex Fe(naphpz)$_3$ in DMSO-d$^6$; Figure S33: ESI-MS spectrum of complex Fe(naphpz)$_3$ in CH$_3$CN; Figure S34: Cyclovoltammetry spectra of complex Fe(naphpz)$_3$ in CH$_3$CN; Figure S35: Plotted data of Randles-Sevcik-Equation Fe(naphpz)$_3$ at different scan rates, first redox step; Figure S36: Plotted data of Randles-Sevcik-Equation Fe(naphpz)$_3$ at different scan rates, second redox step; Figure S37: Plotted data of Randles-Sevcik-Equation Fe(naphpz)$_3$ at different scan rates, first redox step; Figure S38: ATR-IR-spectrum of complex Fe(naphpz)$_3$ in CH$_3$CN; Figure S39: Change in the absorptive behaviour of (Fe(naphpz)$_3$ with an applied potential of 0.8–2 V in CH$_3$CN; Figure S40: Change in the absorptive behaviour of (Fe(naphpz)$_3$ with an applied potential of $-0.5$–($-1.5$) V in CH$_3$CN; Figure S41: Change in the absorptive behaviour of (Fe(naphpz)$_3$ with an applied potential of $-2.5$ V in CH$_3$CN; Figure S42: $^{1}$H-NMR spectrum of complex Fe(MeOppz)$_3$ in CD$_3$CN; Figure S43: $^{13}$C-NMR spectrum of complex Fe(MeOppz)$_3$ in CD$_3$CN; Figure S44: $^{15}$N-HMBC spectrum of Fe(MeOppz)$_3$ in CD$_3$CN; Figure S45: ESI-MS spectrum of complex Fe(MeOppz)$_3$ in CH$_3$CN; Figure S46: Cyclovoltammetry spectra of complex Fe(MeOppz)$_3$ in CH$_3$CN; Figure S47: Plotted data of Randles-Sevcik-Equation Fe(MeOppz)$_3$ at different scan rates, first redox step; Figure S48: Plotted data of Randles-Sevcik-Equation Fe(MeOppz)$_3$ at different scan rates, second redox step; Figure S49: Plotted data of Randles-Sevcik-Equation Fe(MeOppz)$_3$ at different scan rates, third redox step; Figure S50: ATR-IR spectrum of complex Fe(MeOppz)$_3$; Figure S51: Experiments on complex stability in acetonitrile solution: before illumination of Fe(ppz)$_3$ (blue), after illumination of 24 h: Fe(ppz)$_3$ (red), Fe(CF$_3$ppz)$_3$ (green), Fe(MeOppz)$_3$ (purple), Fe(bppz)$_3$ (yellow), Fe)naphpz)$_3$ (orange, pure solubility decreases signal intensity); Figure S52: Experiments on complex stability exemplarily for Fe(ppz)$_3$, bottom blue: before illumination in toluene (blue), from there upwards after illumination: toluene (red), THF (green), DMSO (pruple), DCM (yellow), BuCN (orange), benzene (grey), acetone (red), 2Me-THF (blue); Figure S53: Decomposition of Fe(ppz)$_3$, with different filters. Blue: before irradiation; red: 320 nm bandwidth filter, green: 360 nm band-

width filter, purple: 390 bandwidth filter; yellow: 400 longpass filter, orang: 495 longpassfilter. At 7.90 ppm complex signal, additional diamagnetic species in yellow spectra is the product of the reductive elimination; Figure S54: Calculated slope of the decomposition of $Fe(ppz)_3$, based on the relative intensities of TMS, the product of the reductive illumination at 6.21 ppm and the complex resonance at 7.90 ppm; Figure S55: PBEh-3c optimized lowest quartet state of fac-$Fe(ppz)_3$. *Left*: Spin density plot. *Right*: Depicted Fe-ligand bond lengths; Figure S56: TPSSh/def2-TZVP calculated spatial distribution of the frontier orbitals of the respective complexes; Figure S57: Experimental CtC spectra of $Fe(ppz)_3$, $Fe(bppz)_3$, $Fe(CF_3ppz)_3$ with different substituents; Figure S58: Comparison of experimental and calculated XANES (a,b) and VtC (c,d) spectra with main character of acceptor (a,b) and donor (c,d) orbitals orbital components for Pyrazol (Py) and Phenyl (Ph) accountable for the peak. Table S1: Cyclovoltammetry data for $Fe(ppz)_3$ at different scan rates, first redox step; Table S2: Cyclovoltammetry data for $Fe(ppz)_3$ at different scan rates, second redox step: Table S3: Cyclovoltammetry data for $Fe(ppz)_3$ at different scan rates, third redox step; Table S4: Cyclovoltammetry data for $Fe(bppz)_3$ at different scan rates, first redox step; Table S5: Cyclovoltammetry data for $Fe(bppz)_3$ at different scan rates, second redox step; Table S6: Cyclovoltammetry data for $Fe(bppz)_3$ at different scan rates, third redox step; Table S7: Cyclovoltammetry data for $Fe(CF_3ppz)_3$ at different scan rates, first redox step; Table S8: Cyclovoltammetry data for $Fe(CF_3ppz)_3$ at different scan rates, second redox step; Table S9: Cyclovoltammetry data for $Fe(naphpz)_3$ at different scan rates, first redox step; Table S10: Cyclovoltammetry data for $Fe(naphpz)_3$ at different scan rates, second redox step; Table S11: Cyclovoltammetry data for $Fe(naphpz)_3$ at different scan rates, second redox step; Table S12: Cyclovoltammetry data for $Fe(MeOppz)_3$ at different scan rates, first redox step; Table S13: Cyclovoltammetry data for $Fe(MeOppz)_3$ at different scan rates, second redox step; Table S14: Cyclovoltammetry data for $Fe(MeOppz)_3$ at different scan rates, second redox step; Table S15: Differential Gibbs free energy of PbEh-3c optimized structures excluding and including the SMD model for MeCN and BuCN implemented in ORCA. Negative values account for higher stability; Table S16: Absolute an differential Gibbs free energy of PbEh-3c optimized structures of fac- and mer- isomers; Table S17: Comparison of the bond length and binding angles for the single crystal structure analysis and PBEh-3c geometry optimized fac-complexes. Table S18: Analysis of the main acceptor and donor orbital contribution to the TD-DFT calculated vertical transitions (Figure 5) of fac-$Fe(ppz)_3$. The lettering of the transitions a-h refers to the assigned transitions in Figure 5. Additionally, the calculated wavelength $\lambda$ and oscillator strength $f$ is given for the selected transitions. Figures S1–S50 and Tables S1–S14: Summarized data for complex analysis, including NMR-, IR, MS-spectra, as well as the complete CV data including reversibility determination after Randles-Sevcik; Figures S51–S54: NMR-spectra and caculations of the illumination experiments; Figures S55–S58 and Tables S15–S17: Computational data.

**Author Contributions:** Methodology, synthesis, writing—original draft preparation, T.H.; DFT calculations, writing, L.F.; supporting synthesis, F.L.; writing—review, J.S.; single-crystal measurements and analyses, R.S.; spectroelectrochemistry measurements, A.N.; NMR measurements, H.E.; writing—review and editing, M.B. All authors have read and agreed to the published version of the manuscript.

**Funding:** This research was funded by the German Research Council (Deutsche Forschungsgemeinschaft DFG), in the frame of priority program SPP 2102, grant number BA 4467/7-1 and BA 4467/7-2.

**Data Availability Statement:** Crystallographic data have been deposited at the Cambridge Crystallographic Data Centre and assigned the deposition numbers 2191100-2191104. Copies are available free of charge via www.ccdc.cam.ac.uk, accessed on 1 January 2020.

**Acknowledgments:** The European Synchrotron Radiation Facility (ESRF) and the beamline ID26 are acknowledged for the provision of beamtime and help during the experiment. Generous grants of computer time at the Paderborn Center for Parallel Computing PC2 are gratefully acknowledged. L.F. thanks the chemical industry fund ("Fonds der Chemischen Industrie") for a Kekulé PhD scholarship.

**Conflicts of Interest:** The authors declare no conflict of interest.

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
