# Peer review of "Iron(III)-Complexes with N-Phenylpyrazole-Based Ligands"

_inorganics, doi:10.3390/inorganics11070282_

Round 1
Reviewer 1 Report
This paper describes a series of Iron(III)-Complexes with N-Phenylpyrazole-based Ligands and includes synthesis and crystal structures. Advanced X-ray spectroscopy and DFT studies are used to investigate the electronic properties of the compounds.
The work is well done but the presentation can be significantly improved.
This paper contains 5 crystal structures. The checkcif files shows that the structures have been refined satisfactorily. There are no significant A or B alerts. However the dimensions including H atoms should be included in the cif files.
The description of the structures needs improvement. More details of the refinements should be given including the disorder and how it was treated. The presence of twinning should be mentioned. Also added should be details of the solvent where included and the SQUEEZE treatment whenused.
The numbering system should be simplified. It would be appropriate to have common numbering for the six atoms in the coordination spheres as N1, N2, N3, C1, C2 and C3. To give all atoms 3 digits is just unnecessary. The numbering scheme should be given in Figures. Table S17 is unsatisfactory providing just average dimensions. All dimensions (6 bonds 15 angles ) of the coordination sphere should be included in this . This could be done in a table with five columns for the five structures so that values could be compared. Values for the DFT structures could also be included.
It is particularly important to include all the Fe bond lengths as the authors make claims that distances in Fe(ppz)3 and Fe(CF3-ppz)3 are significantly shorter than in the other molecules which seems unproven given the standard deviations and therefore differences between the similar bonds in all molecules need to be shown.
In the tables of crystallographic information, only the unique cell dimensions are required and fixed angles are unnecessary.
The absolute values given in Table S15 and S16 are not helpful especially when given to an unreasonable 9 digits. Best just to give the relative values in Table S15 compared to the non-solvent and in Table S16 of the fac structures compared to the mer values. Table S18 is incomplete. The caption gives little information. Which structure is being considered? The number of the frontier orbitals should be replaced by identification with reference to HOMO-x and LUMO+x It should be mentioned that the lower case letters reference the peaks in the Table in the text and the intensity of those peaks should be given in Table S16. When the orbital contribution of a transition is only 10% it seems inconsistent not to include other contributions.
While the presentation should be improved, the work done is of good quality and merits publication after revision.
It needs some improvement.
Author Response
This paper describes a series of Iron(III)-Complexes with N-Phenylpyrazole-based Ligands and includes synthesis and crystal structures. Advanced X-ray spectroscopy and DFT studies are used to investigate the electronic properties of the compounds.
The work is well done but the presentation can be significantly improved.
This paper contains 5 crystal structures. The checkcif files shows that the structures have been refined satisfactorily. There are no significant A or B alerts. However the dimensions including H atoms should be included in the cif files.
-> All crystal structures are deposited on CCDC which is an international accepted standard. Since no A or B alerts were obtained, the crystal structures were of high quality. Despite our long term experience in X-ray crystallography, we are a bit puzzled by the expression “dimensions including H atoms”. The cell dimensions are given in the cif-file (and the SI) and we do not understand what the reviewer requests here. If this issue is further specified we are happy to address it.
The description of the structures needs improvement. More details of the refinements should be given including the disorder and how it was treated. The presence of twinning should be mentioned. Also added should be details of the solvent where included and the SQUEEZE treatment whenused.
-> We thank the reviewer for this hint, the requested details were added to the SI.
The numbering system should be simplified. It would be appropriate to have common numbering for the six atoms in the coordination spheres as N1, N2, N3, C1, C2 and C3. To give all atoms 3 digits is just unnecessary. The numbering scheme should be given in Figures.
-> Although we understand the reviewers remark, we kindly disagree. The numbering of the structures is given in the way, that the first digit of the number represents the complex or the ligand. Neighboring atoms of the same element were numbered consecutively in the second or third (if necessary) digit. These measures serve to provide clarity in the atom list. To address this remark partially we added new figures including the atom labels to the SI.
Table S17 is unsatisfactory providing just average dimensions. All dimensions (6 bonds 15 angles ) of the coordination sphere should be included in this . This could be done in a table with five columns for the five structures so that values could be compared. Values for the DFT structures could also be included. It is particularly important to include all the Fe bond lengths as the authors make claims that distances in Fe(ppz)3 and Fe(CF3-ppz)3 are significantly shorter than in the other molecules which seems unproven given the standard deviations and therefore differences between the similar bonds in all molecules need to be shown.
-> We have carefully considered this reasonable criticism and discussed it in the author team. However, with all due respect to the reviewers comment, we felt not comfortable in including all dimensions in this context. Since our complexes exhibits a Δ- and Λ- configuration, one bond length/angle is averaged over 6 values. This would come down to 18 additional values for each of the complexes. Since this would be in conflict with the easy comparison of the values the feviewer so rightfully wanted to emphasize, we would like to refrain from illustrating them in one table. Nevertheless, we want to emphasize, that the deviations within one bond length/angle does only differ marginally, if at all, so that we were only able to comfortably average the values.
In the tables of crystallographic information, only the unique cell dimensions are required and fixed angles are unnecessary.
-> The angles were initially given for non-expert readers but were deleted following your advice.
The absolute values given in Table S15 and S16 are not helpful especially when given to an unreasonable 9 digits. Best just to give the relative values in Table S15 compared to the non-solvent and in Table S16 of the fac structures compared to the mer values. Table S18 is incomplete. The caption gives little information. Which structure is being considered? The number of the frontier orbitals should be replaced by identification with reference to HOMO-x and LUMO+x It should be mentioned that the lower case letters reference the peaks in the Table in the text and the intensity of those peaks should be given in Table S16. When the orbital contribution of a transition is only 10% it seems inconsistent not to include other contributions.
-> We appreciate theses suggestions for improvements in clarity and description of the shown data. We calculated the much more meaningful energy differences in Tab. S15 and Tab. S16. We extend the caption of Tab. S18 with the considered complex fac-Fe(ppz)3 and explained cases a-h to be more precise about the description of the shown information. The absolute numbers of the orbitals are replaced by relative values like HOMO-x/LUMO+x. We added some information like the wavelength and oscillator strength of the calculated transitions and included more contributions to some signals where just contributions of 10% were shown before. Additionally, we extended Fig. 5 in the paper with an analysis of all excited states which is shown as bars in different colors per contribution to exclude the possibility that the signal is dominated by many very small amounts that have not been explicitly analyzed in Tab. S18.
While the presentation should be improved, the work done is of good quality and merits publication after revision.
-> We highly appreciate the comment by the reviewer.
Reviewer 2 Report
The authors present a comprehensive study on complexes of Fe(III) with N-phenylpyrazole ligands. They describe the synthesis, single crystal structures supplemented by X-ray spectroscopy, UV-Vis-spectra, electrochemical study, and DFT calculations. The structures and their optical and electrochemical properties are interesting in point of view of searching for photoactive compounds.
In my opinion this manuscript may be published after some minor revisions.
1. Please show and discuss in the main text at least one example of the HOMO/LUMO orbitals calculated for the studied compounds. Can you combine the theoretically calculated energy gaps with the electrochemical performance. Do they correlate? Can you provide some comparison to other similar structures already published?
2. Please add the description of X-ray single crystal diffractometry experiment and the basic crystallographic data in the main text of the manuscript, not in the supplement only. Do you recognize some correlations between the symmetry of the complexes or the steric hindrance and the electronic behavior?
3. The authors declare that the “single-crystals suitable for X-ray diffraction were obtained by diffusion of cyclopentane into a solution of the respective complex in DCM”. My question is what was the structure of the materials used for other experiments? Please perform an X-ray powder diffraction experiments on the samples obtained directly after the synthesis and compare the diffractograms with the single crystal X-ray structure. Were the samples pure, had they the same structures?
4. Have you considered to combine your complexes with some other materials? Such hybrid systems may enhance the performance of photoelectric effect in co-sensitized solar cells. Please see for example here (DOI 10.3390/ijms231710005,DOI 10.1039/c4dt03602f).
Author Response
The authors present a comprehensive study on complexes of iron(III) with N-phenylpyrazole ligands. They describe the synthesis, single crystal structures supplemented by X-ray spectroscopy, UV-Vis-spectra, electrochemical study, and DFT calculations. The structures and their optical and electrochemical properties are interesting in point of view of searching for photoactive compounds.
In my opinion this manuscript may be published after some minor revisions.
-> We highly appreciate this positive comment by the reviewer.
- Please show and discuss in the main text at least one example of the HOMO/LUMO orbitals calculated for the studied compounds. Can you combine the theoretically calculated energy gaps with the electrochemical performance. Do they correlate? Can you provide some comparison to other similar structures already published?
-> We thank the reviewer for the suggestion to correlate the energies with the experimental electrochemical data, as these show a very good agreement. We added an exteded discussion of the HOMO and LUMO (b) energies but none about the gap because it is small (same orbital set) and finds no correspondence to experimental data. The HOMO and LUMO gap in the a orbital set would correlate to the MLCT energy, but no ligand reduction could be obtained in the CV. However, a static LMCT energy can be extracted from the CV potentials, so we correlated this with the calculated orbital energies of the orbitals involved in the LMCT. Since the discussed beta HOMO and LUMO orbital both have the described metal character an illustration would offer little additional information, but a link to the SI in which they are shown is made. Comparison of already published structures is difficult, since neither is data for similar compounds available, nor a series of functional groups. Only the Co(ppz)3 might be mentionable in this context, but we refrained from doing so, since the comparison between d6 and d5 complexes seems inappropriate.
- Please add the description of X-ray single crystal diffractometry experiment and the basic crystallographic data in the main text of the manuscript, not in the supplement only. Do you recognize some correlations between the symmetry of the complexes or the steric hindrance and the electronic behavior?
-> The reviewer has made some excellent points and we sincerely appreciate this well-thought comment. We added the description of the Single Crystal X-Ray diffractometry in the main text under “Materials and Methods”. We do believe that the steric hindrance or the symmetry may be possible, but unlikely to have a huge influence on the electronic behavior. We therefore concentrated the discussion on the functional groups attached to the ligand motif.
- The authors declare that the “single-crystals suitable for X-ray diffraction were obtained by diffusion of cyclopentane into a solution of the respective complex in DCM”. My question is what was the structure of the materials used for other experiments? Please perform an X-ray powder diffraction experiment on the samples obtained directly after the synthesis and compare the diffractograms with the single crystal X-ray structure. Were the samples pure, had they the same structures?
-> At this stage, we agree with Reviewer 2 that the complex structure should be consistent throughout all studies. Unfortunately, we were unable to address this issue using X-Ray powder diffraction and had to resort to other methods. We performed NMR analyses after crystallization, from which we obtained single crystals for X-ray diffraction. These experiments prove, by the set of obtained resonances, that only fac-isomers are present. We repeated the experiments after the complex was worked up before crystallization and obtained the same set of resonances. We were also able to crystallize Fe(ppz)3 in a variety of solvent combinations. Again, we observed the fac-isomer as exclusive product in the NMR. Furthermore, comparing mass spectrometry and elemental tests before and after crystallization demonstrated that we were able to obtain pure products in both circumstances. We were able to validate the stability of the fac-isomer for all solvents examined in this research using NMR spectroscopy. Therefore, it can be excluded, that the structures determined by single crystal X-ray diffraction differ from the structure in solution.
- Have you considered to combine your complexes with some other materials? Such hybrid systems may enhance the performance of photoelectric effect in co-sensitized solar cells. Please see for example here (DOI 10.3390/ijms231710005,DOI 10.1039/c4dt03602f).
-> We thank the reviewer for this suggestion; we have not yet investigated dye sensitized solar cells etc yet. We will address this possibility in the future, but it is not within the scope of the present study.
Reviewer 3 Report
Tanja and co-workers present a paper on an in dept characterization of a series of meta substituted 1-phenylpyrazolato complexes of Fe(III) for photochemical application. The tecniques presented are well described and analized, making possible to the reader to deeply understand the structure and the physical phenomena in ppresence of light, also through a well developed modelling of the exited states. The crystallographic part is expecially well done, with and in dept interpretation of the electronic effects that changes the relative M-L distances and the preference of fac isomer for all the presented compounds. The language is correct and the reading is easy, making this article very nice. My only curiosity is on the reactions: with these very low yeald, how are scalable these reactions? How much product you manage to sinthetize? Have you an idea of the nature of the enormous ammount of byproducts?
For this reasons I'm quite happy to say that this article can be accepted by Inorganics after minor revision
Author Response
Tanja and co-workers present a paper on an in dept characterization of a series of meta substituted 1-phenylpyrazolato complexes of Fe(III) for photochemical application. The tecniques presented are well described and analized, making possible to the reader to deeply understand the structure and the physical phenomena in ppresence of light, also through a well developed modelling of the exited states. The crystallographic part is expecially well done, with and in dept interpretation of the electronic effects that changes the relative M-L distances and the preference of fac isomer for all the presented compounds. The language is correct and the reading is easy, making this article very nice. My only curiosity is on the reactions: with these very low yeald, how are scalable these reactions? How much product you manage to sinthetize? Have you an idea of the nature of the enormous ammount of byproducts?
For this reasons I'm quite happy to say that this article can be accepted by Inorganics after minor revision.
-> We highly appreciate the reviewer's positive words about our article. We share your curiosity about our low yield. As stated in our work, we believe that the formation of undercoordinated products and decomposition effects lead to undesired by-products. Additionally the usage of Grignard reagents might cause the formation of other products which we did not characterize further. Deprotonation in positions other than the ortho cannot be completely ruled out.
Reviewer 4 Report
Matthias Bauer and coworkers have reported an important story on iron(III)-complexes with N-Phenylpyrazole-based Ligands. They have synthesized five complexes using the tris(1-phenylpyrazolato-N,C²)iron(III) complex in alternative to the expensive noble metal complexes. Although there are reports of similar compounds, the authors have done extensive studies with many experiments such as XRD, CV, optical spectroscopy (UV-Vis), computational calculations, hard XRD spectroscopy, etc. They have characterized and studied their properties in an extensive and informative way. I would say that paper deserved to be accepted for publication in inorganics.
However, I would like the author to change a little bit and provide some more information.
(1) Please write the name of the iron complexes in the captions as they are shown in Figure 1.
(2) Please mention why the yield of Fe(MeOppz)3, Fe(bppz)3, Fe(naphpz)3 and Fe(ppz)3 are so low.
(3) Please mention the role of EtMgBr in the reaction (if possible, with the mechanism).
(4) Crystal structures of the compounds can be shifted to the main text in the main text.
(5) The GOOF values of two compounds 1.311 (CCDC number 2191103) and 1.143 (CCDC number 2191104) can be reduced.
Author Response
Matthias Bauer and coworkers have reported an important story on iron(III)-complexes with N-Phenylpyrazole-based Ligands. They have synthesized five complexes using the tris(1-phenylpyrazolato-N,C²)iron(III) complex in alternative to the expensive noble metal complexes. Although there are reports of similar compounds, the authors have done extensive studies with many experiments such as XRD, CV, optical spectroscopy (UV-Vis), computational calculations, hard XRD spectroscopy, etc. They have characterized and studied their properties in an extensive and informative way. I would say that paper deserved to be accepted for publication in inorganics.
However, I would like the author to change a little bit and provide some more information.
(1) Please write the name of the iron complexes in the captions as they are shown in Figure 1.
-> We apologize for the missing names of the complexes within the main text. The new caption reads as follows:
Figure 1: Top: Reaction pathway for the synthesis of pyrazole-based iron(III) complexes, exemplary for tris(1-phenylpyrazolato-N,C2′)iron(III) (Fe(ppz)3). Bottom: Structures of tris(1-(4-(trifluoromethyl)phenyl)pyrazolato-N,C²)iron(III) (Fe(CF3ppz)3), tris(1-(([1,1'-biphenyl])-4-yl)phenyl)pyrazolato-N,C²)iron(III) (Fe(bppz)3), tris(1-(naphthalen-2-yl)pyrazolato-N,C²)iron(III) Fe(naphpz)3, tris(1-(4-methoxyphenyl)pyrazolato-N,C²)iron(III) (Fe(MeOppz)3), with their respective yields.
(2) Please mention why the yield of Fe(MeOppz)3, Fe(bppz)3, Fe(naphpz)3 and Fe(ppz)3 are so low.
-> As already mentioned in the manuscript, we believe that the disproportion reaction to create iron(III) and undesirable by-products is the primary cause of our low yields. These may also occur as a result of the minimal acidity difference between protons in the phenyl-pyrazole motif. Deprotonation in position other than the desired ortho-position cannot be completely ruled out. Unfortunately, alternative bases/gignards either produced the same result or produced no complex at all.
(3) Please mention the role of EtMgBr in the reaction (if possible, with the mechanism).
-> We appreciate this suggestion and added it to figure 1.
(4) Crystal structures of the compounds can be shifted to the main text in the main text.
-> We thank the reviewer for this suggestion. However we would kindly refrain from following it, as we believe that the crystal structures in the main text disrupt the story line. Therefore, we put them in the "Materials and Methods" section of the paper, within the analyses of each complex. We cannot remove the second image in this section since we want to emphasize the specific position of NMR resonances to the matching place in the structure.
(5) The GOOF values of two compounds 1.311 (CCDC number 2191103) and 1.143 (CCDC number 2191104) can be reduced.
-> We indeed tried to obtain the very best fitting values for the presented structures and to the best of our knowledge, the GooF cannot be reduced furthermore. If the reviewer has an advice how to optimize the refinement, we would be more than happy to apply it to our data.